# Fibrodysplasia Ossificans Progressiva: A Challenging Diagnosis

**DOI:** 10.3390/genes12081187

**Published:** 2021-07-30

**Authors:** Daniele De Brasi, Francesca Orlando, Valeria Gaeta, Maria De Liso, Fabio Acquaviva, Luigi Martemucci, Augusto Mastrominico, Maja Di Rocco

**Affiliations:** 1Department of Pediatrics, AORN Santobono-Pausilipon, 80122 Naples, Italy; francesca.orlando8@gmail.com (F.O.); val.gaeta1992@gmail.com (V.G.); fabio.acquaviv@gmail.com (F.A.); l.martemucci230@gmail.com (L.M.); augustomastrominico@alice.it (A.M.); 2Department of Translational Medicine, “Federico II” University of Naples, 80131 Naples, Italy; 3Department of Neurosciences and Rehabilitation, AORN Santobono-Pausilipon, 80122 Naples, Italy; maria.delisomd@gmail.com; 4Department of Pediatrics, IRCCS Istituto “Giannina Gaslini”, 16147 Genoa, Italy; majadirocco@gaslini.org

**Keywords:** fibrodysplasia ossificans progressiva, FOP, *ACVR1*, *ACVR1/ALK2*, heterotopic ossification, MRI, CNS

## Abstract

Fibrodysplasia ossificans progressiva (FOP) is an ultrarare genetic condition characterized by extraskeletal bone formation. Most of the musculoskeletal characteristics of FOP are related to dysregulated chondrogenesis, with heterotopic ossification being the most typical feature. Activating mutations of activin receptor A type I (ACVR1), a bone morphogenetic protein (BMP) type I receptor, are responsible for the skeletal and nonskeletal features. The clinical phenotype is always consistent, with congenital bilateral hallux valgus malformation and early-onset heterotopic ossification occurring spontaneously or, more frequently, precipitated by trauma. Painful, recurrent soft-tissue swellings (flare-ups) precede localized heterotopic ossification that can occur at any location, typically affecting regions near the axial skeleton and later progressing to the appendicular bones. A diagnosis of FOP is suspected in a proband presenting with hallux valgus malformation, heterotopic ossification, and confirmed by the identification of a heterozygous pathogenic variant in the *ACVR1/ALK2* gene. Avoiding unnecessary surgical procedures, prescribing prophylactic corticosteroids, preventing falls, and using protective headgear represent essential interventions for care management. Different classes of medications to contain acute inflammation flare-ups have been proposed, with high dose corticosteroids and nonsteroidal anti-inflammatory drugs usually utilized. Here, we report on two FOP patients, with typical clinical features summarizing the principal aspects of FOP, and we aim to provide comprehensive information outlining some unusual findings, possibly contributing to FOP’s definition and management.

## 1. Introduction

Fibrodysplasia ossificans progressiva (FOP, OMIM #135100) is a very rare disabling autosomal dominant genetic disorder, characterized by progressive heterotopic endochondral ossification (HEO). First described in 1740 in the Philosophical Transaction of the Royal Society of London by John Freke, FOP was named one century later myositis ossificans progressive (MOP) because of the episodic inflammatory flare-ups involving skeletal muscle. The name fibrodysplasia ossificans progressiva (FOP) was proposed by Bauer and Bode in 1940, then adopted in 1960 by McKusick for the primary connective tissue involvement of tendons, ligaments, fascia, and aponeuroses [1]. 

The molecular basis of FOP was identified in 2006, when activating mutations in activin receptor A type I gene (*ACVR1*, also referred to as activin receptor-like kinase 2, *ALK2*), which codes for a bone morphogenetic protein (BMP) type I receptor, were recognized as the proximate genetic cause of FOP in all affected individuals [2]. The *ACVR1/ALK2* gene has been mapped to chromosome region 2q23q24 by linkage analysis [2] and by fluorescence *in situ* hybridization [3]. Most patients presenting with the classic FOP phenotype of progressive heterotopic ossification (HO) and great toe malformation share the same heterozygous mutation in the *ACVR1/ALK2* gene (c.617G>A; R206H); this mutation is identified in more than 97% of affected individuals. Other pathogenic variants of the *ACVR1/ALK2* gene have been more recently identified, all located in the glycine-serine (GS)-rich domain or in the protein kinase domain [4] (Figure 1). Both the *ACVR1*^R206H^ mutation and all the variants reportedly show mild constitutive activity and in vitro enhanced ligand-dependent activity of BMP signaling. Dysregulation of the BMP signaling pathway is responsible for modification of osteochondrogenesis that represents the key point of the musculoskeletal phenotype of FOP [5]. A possible model of BMP signaling pathway disruption in FOP proposes that BMPs bind to complexes of type I and type II serine/threonine kinase BMP receptors (such as ACVR1) on the cell surface to activate intracellular signal transduction via R-SMADs SMAD1/5/9(8). Phosphorylated R-SMADs form a complex with the comediator SMAD4, which translocates into the nucleus, and regulates transcription that drives endochondral ossification. Interestingly, recent studies established the role of inflammation in HO genesis and propagation in FOP patients. Specifically, it has been shown that activin A, which is expressed by innate immune system cells, plays an important role in both promoting and resolving inflammation, particularly by blocking ACVR1^WT^ signaling. Activin A is effectively perceived as a BMP ligand by ACVR1^R206H^ leading to downstream BMP signaling via SMAD1/5/9(8), thus responding to activins A, AB, AC, and B, to which the wild-type ACVR1 is unresponsive [6].

The penetrance of the *ACVR1/ALK2* gene mutations is complete, and they usually arise de novo [7]. 

Based on several population studies, the prevalence of FOP is estimated at approximately 1 in 1 million (0.6–1.36:1,000,000), with no racial, ethnic, gender, or geographic predispositions identified [8].

Although the FOP phenotype is extremely consistent and easily recognizable when fully expressed, early diagnosis still presents a challenge to physicians. According to the registry of the International FOP Association (IFOPA), the mean age at which the first symptoms occur is 5.4 years, while the mean age of FOP diagnosis is 7.5 years [6]. Almost all patients with FOP are typically born with various types and degrees of congenital great toe malformations, such as bilateral hallux valgus malformation with the lack of a toe crease at the metatarsophalangeal joints, and, in some cases, macrodactyly. Early-onset spontaneous or trauma precipitated HO represents the hallmark of the syndrome. Signs of HO start to occur episodically in the first decade of life. Recurrent painful soft-tissue swellings (flare-ups) may precede localized heterotopic ossification, which can occur at any location but typically affect regions in proximity to the axial skeleton in the early/mild stages before progressing to the appendicular bones, eventually leading to restriction of movement due to ossification impacting joint mobility. Flare-ups are generally sporadic and unpredictable and are caused by the underlying inflammation in the ligaments, tendons or skeletal muscles occurring upon pro-inflammatory insults such as muscle fatigue, tissue damage, intramuscular injections, or viral illness. Over time, because of repeated flare-ups at different sites, progressive and cumulative ossification of soft tissues occurs, resulting in ankylosis of various joints and leading to the debilitating effects of FOP. Most patients usually become wheelchair-bound by the end of the second decade. Some patients may experience hearing loss due to middle ear ossification. The involvement of the jaw can lead to swallowing and feeding difficulties, resulting in malnourishment and gradual weight loss. Speaking can also be involved. Debilitating effects of FOP start to become life-threatening when involvement of the intercostal muscles, costovertebral joints and thoracic paravertebral soft tissue occurs, resulting in thoracic insufficiency syndrome. Patients eventually die because of cardiorespiratory complications, such as pneumonia or right-sided heart failure, and have a median life span of around 40 years [6].

According to clinical presentation, patients with the classic FOP phenotype, who represent about 92% of cases and all have the canonical *ACVR1/ALK2* gene c.617G>A (p.R206H) mutation, show both the typical congenital malformations of the great toe and progressive HO with other common but variable features of FOP such as tibial osteochondromas, conductive hearing impairment, age-depended increased risk of hypercalciuria-related nephrolithiasis, sparse hair and eyebrows, and neuroimaging abnormalities especially involving the pontine region [6]. Additional findings, often related to specific *ACVR1/ALK2* mutations, are summarized in Table 1.

There are no formal diagnostic criteria for FOP. An FOP diagnosis is suspected in a proband with hallux malformation and heterotopic ossification and is confirmed by the presence of a heterozygous pathogenic variant in the *ACVR1/ALK2* gene. Otherwise, FOP should be suspected in individuals with the following clinical and radiographic findings: congenital hallux valgus deformity or other hallux malformations, most often bilateral; progressive heterotopic ossification manifesting as a palpable mass, either spontaneous or in response to soft-tissue trauma; and painful, recurrent soft-tissue swelling (flare-ups) occurring in the form of scalp nodules in infancy. It should be kept in mind that a lack of suspicion leads to delayed diagnosis or misdiagnosis and, in most cases, to inappropriate and unnecessary testing. In these cases, invasive biopsies may cause flare-ups, promote HO, and cause permanent harm and lifelong disability in as many as 50% of cases.

Hereditary multiple osteochondromas (HMO), progressive osseous heteroplasia (POH), metachondromatosis (METCDS), and brachydactyly type B1 (BDB1) are differential diagnoses for FOP lacking hallux malformations (HMO, POH and METCDS), or heterotopic ossification (HMO, METCDS and BDB1). It should be also considered that hallux anomalies may represent isolated congenital malformations or juvenile bunions, and tumor-like swellings, possibly associated with sarcomas, desmoid tumors, aggressive juvenile fibromatosis, or lymphedema [7]. 

Imaging is useful for FOP diagnosis, but it should be used appropriately. Whereas plain radiographs can only detect skeletal anomalies when present and/or heterotopic ossification [9], the use of magnetic resonance imaging (MRI) can identify muscle pre-osseous inflammatory lesions, avoiding additional diagnostic procedures (including biopsy), leading to the correct genetic diagnosis, and possibly suggesting specific therapeutic interventions [10]. CNS findings on MRI studies have been less investigated, and reveal the presence of CNS anomalies, namely structural malformations, demyelinated lesions, or focal inflammatory changes [11,12,13]. Although biochemical and radiological investigations can provide useful information on the disease process, ultimately diagnosis can only be confirmed by DNA sequence analysis to trace the underlying mutation [7].

Avoiding intramuscular injections and arterial punctures, biopsies, elective surgical procedures, and removal of heterotopic bone is recommended, especially when FOP diagnosis is only hypothesized. Other opportune interventions are fall prevention, the use of protective headgear, and dietetic care for patients with feeding difficulties, and the use of adequate hydration and avoidance of high protein and high salt intake to prevent renal stones are recommended. Avoiding contact sports, overstretching of soft tissues, muscle fatigue and passive range of motion, and supplemental oxygen (which can suppress respiratory drive in individuals with thoracic insufficiency syndrome), are mandatory. Psychological support is recommended for patients and their families [7].

Surveillance should be principally ensured by: annual clinical orthopedics evaluation, annual nutrition evaluation, and examination for jaw ankylosis; baseline pulmonary function assessment, sleep assessment, and echocardiogram before the age of ten years, followed by annual clinical evaluation of respiratory status; annual evaluation of fracture risk; audiology assessment every 12 to 24 months; annual assessment for signs and symptoms of nephrocalcinosis, gastrointestinal complications, and skin integrity; dental examinations every six months; and Doppler ultrasound if deep venous thrombosis is suspected [7]. 

There is no successful treatment for FOP, but some drugs can be used to alleviate the initial symptoms. Corticosteroids are most effective if used within the first 24 h of a flare-up affecting the major joints of the appendicular skeleton or the submandibular area and jaw; the recommended dose of prednisone for acute flare-ups is 2 mg/kg/day (up to 100 mg) as a single daily dose for 4 days, followed by further 10 days symptoms persist. Prednisone—1–2 mg/kg, (orally) once daily for 3–4 days—is often used in order to prevent flare-ups after severe soft-tissue injury and dental or surgical procedures. It can be also used for brief periods in acute joint and periarticular pain. When corticosteroids are not indicated or discontinued, non-steroidal anti-inflammatory drugs (NSAIDs) or selective cyclooxygenase-2 (COX2) inhibitors may be used for the duration of the flare-ups, although there is no evidence that chronic treatment with these drugs prevents flare-ups in FOP. Additional treatments have been used experimentally and anecdotally in the symptomatic management of flare-ups in FOP, although concrete clinical data for these treatments are sparse and it remains unclear if they contributed to decrease pain and/or swelling (Figure 2) [14,15,16]. In addition, at present, some novel pharmacological drugs are being studied, and various clinical trials on FOP are in progress or recruiting participants [15,16,17,18,19,20,21,22,23,24,25] (Figure 3). This new class of medications (referred as class III elsewhere [16]) includes drugs under development and/or currently being tested in clinical trials. In detail, ACVR1/ALK2 signal transduction inhibitors (STI), namely Saracatinib (AstraZeneca; STOPFOP Investigators), IPN60130 (Ipsen), BCX9250 (Bio Cryst), INCB000928 (Incyte) and Ker-047 (Keros), act to block *ACVR1/ALK2* signal transduction, along with the mTOR Inhibitor Rapamycin (Kyoto University), which acts to inhibit *ACVR1/ALK2* signal transduction. In addition, a monoclonal antibody against *ACVR1/ALK2* DS-6016 (Daiichi Sankyo) blocks *ACVR1/ALK2* at the cell surface, whereas the monoclonal antibody against activin A Garetosmab (Regeneron) blocks Activin A signaling through mutant *ACVR1/ALK2*. Finally, the retinoic acid receptor-gamma agonist Palovarotene (Ipsen) inhibits ectopic chondrogenesis [15,16,17,18,19,20,21,22,23,24,25].

Concerning genetic counseling, FOP is inherited in an autosomal dominant manner. Most affected individuals present simplex cases resulting from a de novo *ACVR1/ALK2* gene pathogenic variant. Rarely, an individual affected by FOP has an affected parent. As to prenatal testing, fetal ultrasound may identify a hallux valgus deformity as early as gestational week 23 [26]. If an *ACVR1/ALK2* gene pathogenic variant is identified in an affected family member, prenatal testing for a pregnancy at increased risk and preimplantation genetic testing is feasible [6].

## 2. Clinical Report

*Patient 1* was a 9-year-old female, the second child born to non-consanguineous healthy parents. The pregnancy was uneventful and delivery was by casearean section. The birth weight was 3090 kg. On the 25th day, she was admitted because of bronchiolitis. She then underwent regularly scheduled first-year vaccinations, and an MPR dose at 15 months. Several months later, she developed a hard, slightly warm non-painful swelling, in the right latero-cervical/scapular region following a low-intensity trauma. A new swelling lesion later developed in the dorsal region, bilaterally, with consensual inguinal lymphadenopathy. When admitted at 2 years of age, an accurate physical evaluation was performed, in addition to the swelling lesions, the presence of bilateral hallux valgus deformity, two frontal café-au-lait spots, and a lumbar angioma. A soft-tissue MRI was performed, revealing a hyperintense signal on the right latissimus dorsi and teres major and on the left side of posterolateral muscles of the neck, subscapularis, and infraspinatus muscles, with postcontrast enhancement. A foot X-ray showed abnormalities of the great toe (mild hallux valgus, triphalangism, and accessory distal epiphyseal nucleus) (findings and figures previously reported in [27]). No ectopic ossification was found by CT scan. Cardiologic and abdominal ultrasonography and chest roentgenogram were normal. In the laboratory evaluation, the urinalysis, total blood count, inflammatory markers, infective screening, kidney and liver function, lipidic profile, complement, immunity and autoimmunity profile, and clothing evaluation were all within the normal range. Because of a suspected progressive pyomyositis/myofibromatosis, fine needle ago biopsy (FNAB) from the swallowed lesion in the right scapular region was performed, showing proliferation of bland spindle cells of the fibroblastic/myofibroblastic type, organized in short bundles and fascicles, with loosely textured ground substance and myxoid and the presence of numerous not atypical mitotic figures (findings and figures previously reported in [27]). Subsequently, on the basis of halluces malformation and flare-ups, a diagnosis of FOP was suspected, and sequencing of the *ACVR1/ALK2* gene in DNA from a peripheral blood sample was performed, identifying a de novo recurrent c.617G>A (p.R206H) mutation, thus confirming the diagnosis of FOP. 

*Patient 2* was a 3-year-old boy referred to our hospital because of macrocephaly and frontal painful edema. The child was a full-term male born to unrelated parents. His parents reported an accidental cranial trauma one month before hospitalization, followed by an increase in head circumference some days later and subsequently an initial painful soft mass in the eyelid region. Suspecting urticaria/angioedema due to an insect bite, oral therapy with cetirizine and betamethasone was started without improvement. Because of worsening clinical signs, the patient was admitted to our hospital, where facial painful lesions and macrocephaly were evidenced (occipital-frontal circumference 54 cm, >95th percentile). Blood analysis and urinalysis, assessment of the tumor markers CEA and alpha-fetoprotein, autoimmunity tests and infective screening were performed, all within the normal range, except for an elevated C-reactive protein (CRP) value of 52.76 mg/L (normal range 0–5 mg/L). Cardiologic, neurologic, onco-hematologic, otorhinolaryngologic, and ophthalmic assessments did not reveal any note worthy alterations. A cerebral computed tomography (CT) scan showed the presence of pericranial soft tissue swelling and edema surrounding the cranial bone, prominence of vascular structures, small areas of heterotopic calcifications in the lenticular nuclei, bilateral thickening of the optic nerves, pontine dysmorphism with bulging of the dorsal surface and enlargement of the subarachnoid cisterns anteriorly to brainstem (Figure 4). During hospitalization, the patient continued oral therapy with cetirizine and betamethasone with progressive edema reduction. Several weeks later, a small soft mass appeared in the left cervical region, which progressively expanded to the scapular region and was associated with pain and limited neck motion. Doppler ultrasound examination indicated moderate thickening of all muscles of the posterior cervical and dorsal region (particularly on the left side), characterized by heterogeneous internal echogenicity but without fibrillar structure alterations, and a significant increase in vascularization. Brain MRI revealed pontine dysmorphism with fusion of the facial colliculi and bulging of the dorsal surface protruding into the fourth ventricle, determining a reduction of its dimension; thickened inferior cerebellar peduncles; squared frontal horns; and mild ectasia of the Virchow-Robin spaces, cisterns, and subarachnoid spaces. It also showed remarkable T2/FLAIR hyperintensity in the bilateral dentate nuclei and the adjacent peridentate area of white matter (Figure 5). STIR-coronal and postcontrast T1 MRI of the cervicothoracic region showed a diffuse hyperintense signal and contrast enhancement extending along the muscles and fascial planes of the neck and thoracic wall (Figure 6). Cranial and facial edema progressively extended down the neck, producing a latero-cervical hard swelling, further expanding to the dorsal region in the sub-scapular area. Genetic counseling raveled the presence of a proximal implant of the big toe, shorter than the second toe and with bilateral valgus deviation, in addition to the neck and dorsal lesions. A diagnosis of FOP was hypothesized. Molecular analysis of DNA from a peripheral blood sample identified a *de novo* heterozygous mutation in the *ACVR1/ALK2* gene (c.617A>G, p.R206H), confirming the clinical suspicion.

Both patients were referred to the Italian Center of Reference for FOP and they are currently in follow-up with sporadic episodes of flare-ups briefly managed with a bolus of low-dose steroids.

## 3. Discussion

FOP is an ultra-rare condition, and its precocious diagnosis is essential to avoid inappropriate procedures that could worsen and accelerate a clinically unfavorable course and to accomplish the correct management using appropriate preventive and therapeutic strategies. Although it is recognizable and easy to suspect, an FOP diagnosis is usually reached at a mean age of 7.5 years [6]. The cases reported here received precocious diagnoses, at 2.5 and 3 years of age, far before the usual age of diagnosis. Although the first patient underwent FNAB because of a suspected progressive pyomyositis/myofibromatosis, a precocious diagnosis was made in both cases, allowing for correct management for flare-ups prevention and therapeutic intervention. In this respect, the first major issue to consider is that clinicians should try to diagnose FOP as early as possible. As the only clinical sign present before flare-ups (i.e., in asymptomatic patients) could be hallux malformations, usually evident at birth in affected patients, it is reasonable to recommend that clinicians be aware of its presence during patient evaluations, especially when unusual and unexplained swelling lesions occur. In fact, although hallux malformation is a relatively frequent finding often isolated or presenting as a part of a brachydactylic phenotype, it should always alert clinicians in suspecting FOP, especially when no family history of foot malformations are noted.

Case 2 presented with macrocrania as the first early clinical sign. His cranic circumference progressively increased some days after an accidental cranial trauma, followed by the appearance of a painful soft mass in the eyelid region, with no neuroimaging signs of intracranial lesions. This unusual onset is rarely reported in the literature (macrocephaly is usually reported as a congenital finding in some patients) and may be considered as a possible not infrequent precocious sign of FOP, as cranial trauma is not so rare in infancy and childhood [16]. Furthermore, this finding should alert clinicians after head trauma in childhood, when macrocrania is secondary to pericranial swelling.

Imaging contributes greatly to FOP diagnosis. Whereas MRI systemic findings, including muscle and skeleton features, are well-known, CNS MRI findings are less recognized, especially in childhood [11,12,13]. Patient 2 in the present report received a CNS MRI assessment, revealing a pontine anomaly and fusion of the facial colliculi, determining a reduction of the fourth ventricle’s dimension, thickened inferior cerebellar peduncles, squared frontal horns, and mild ectasia of the Virchow–Robin and subarachnoid spaces. The assessment also showed remarkable T2/FLAIR hyperintensity, in bilateral dentate nuclei and the adjacent cerebellar white matter (see Figure 4). In 2012, Kan et al. [12] described CNS MRI findings of four patients (two adults and two children), detecting extensive hyperintense lesions in the cerebral white matter, the spinal cord, the dorsal pons and the dentate nuclei bilaterally and surrounding the fourth ventricle. These findings were similar to those described in patient 2. Furthermore, similar lesions with demyelination and focal inflammatory changes were also consistently observed in two transgenic knock-in mouse models, suggesting that dysregulated BMP signaling disturbs the normal homeostasis of target tissues, including the CNS. On the other hand, Severino et al. [10] performed an observational cross-sectional brain MRI study in 13 patients and evidenced that all patients presented small asymptomatic lesions resembling hamartomas at the level of the dorsal medulla and ventral pons, which were associated with minor brainstem dysmorphisms and other CNS abnormalities. Patient 2 had similar pons lesions, resembling hamartomas. Furthermore, he did not present any clinical CNS signs/symptoms to date. These observations further support the hypothesis that the effects of mutations of the *ACVR1/ALK2* gene also extend to the CNS, and these CNS lesions could be considered a benign process, because of the absence of corresponding clinical symptoms.

It is note worthy that some somatic recurrent constitutively activating mutations in the *ACVR1/ALK2* gene (namely R258G, R206H, G356D, and G328E/V/W), previously described in FOP patients, have been reported in 21% of diffuse intrinsic pontine glioma (DIPG) samples [28,29]. Although patient 1 did not receive any neuroimaging investigation, she did not present any neurologic symptoms suggesting DIPG diagnosis; on the other hand, patient 2 did not show any CNS images of DIPG on the MRI. However, although FOP and a group of DIPGs share the same *ACVR1/ALK2* gene mutations, germline and somatic respectively, no cases of DIPG have been described in FOP patients to date, probably because *ALK2* mutations likely do not initiate tumorigenesis themselves, but rather lead to DIPG when coexisting with other mutations. Nevertheless, growing evidence suggests that *ALK2* represents a therapeutic target for both conditions [30].

Both patients were occasionally treated with corticosteroids, corresponding to flare-ups or major joint pain, according to validated guidelines. No other therapeutic approaches were used, as corticosteroid treatment appears to be a more effective and safer treatment option, especially when no persistent symptoms are observed. Many other treatments have been used in FOP management, but none appear to be absolutely effective. Mast cells could play an important role in the pathology of heterotopic ossification in FOP, so Imatinib, a tyrosine kinase inhibitor initially developed for chronic myeloid leukemia that has antiproliferative and immunomodulatory effects in these cells has been used to mitigate severe and unrelenting flare-ups for FOP. However, data on the real and long term efficacy of Imatinib in FOP have yet to be clarified [14]. Bisphosphonates have been used experimentally and anecdotally in the symptomatic management of flare-ups in FOP, although concrete clinical data for these treatments are sparse and it remains unclear whether bisphosphonates affect flare-ups [16]. Intravenous aminobisphosphonates are indicated for the prevention and treatment of steroid-associated bone loss, a common problem in FOP patients. The chronic use of antiangiogenic agents, calcium binders, colchicine, fluoroquinolone antibiotics, mineralization inhibitors, PPAR-gamma agonists, TNF-α inhibitors and warfarin have been described anecdotally or reported with either unsatisfactory or equivocal results [16]. The use of these medications has still not even validated. On the other hand, increasing clinical trials on FOP are in progress or recruiting participants. Most appear promising, but only a few are in phase 2 or 3, so their efficacy must still be demonstrated [15,16,17,18,19,20,21,22,23,24,25].

## 4. Conclusions

Diagnoses of FOP are often overlooked or delayed, thus resulting in possible inopportune testing (including biopsies and other surgical interventions), which could worsen clinical course and lead to poor patient management. In order to alert clinicians, we summarize, according to the most recent fuidelines on FOP [16], the following pivotal recommendations to manage FOP patients appropriately:Diagnosis of FOP is clinical (skeletal malformations including malformed great toes and soft tissue swelling and progressive HO) but requires genetic confirmation (presence of *ACVR1/ALK2* gene mutation).If FOP is suspected, all elective procedures such as surgeries, biopsies, intramuscular injections, and immunizations should be deferred until a definitive diagnosis is made.Fall prevention and the use of protective headgear are recommended.Avoiding contact sports, overstretching of soft tissues, and muscle fatigue are strongly recommended.Specific surveillance should be ensured in FOP patients, and each patient should have a primary physician who consults with an FOP expert and helps to coordinate a local care team.Although no effective treatment is available for FOP, some drugs (namely corticosteroids, NSAIDs, and COX2 inhibitor) are approved for FOP patients to relieve initial symptoms and flare-ups, thus improving their quality of life; a growing number of treatments is progressively used/studied, including a consistent group of drugs under clinical trials.

Because of its complexity, a diagnosis of FOP remains a challenging for clinicians.

## Figures and Tables

**Figure 1 genes-12-01187-f001:**
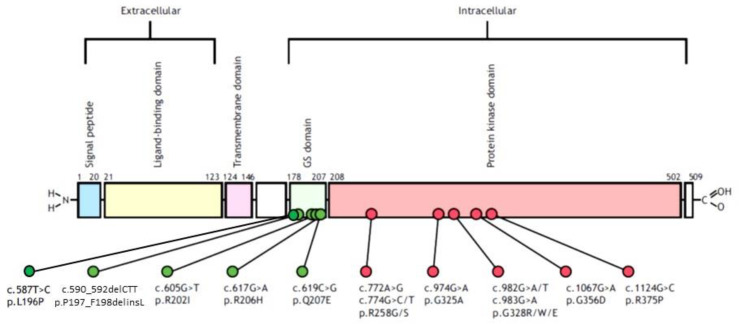
Updated schematic representation of human ACVR, indicating domains and locations of mutations of the *ACVR1/ALK2* gene. GS, glycine-serine (from [6], modified).

**Figure 2 genes-12-01187-f002:**
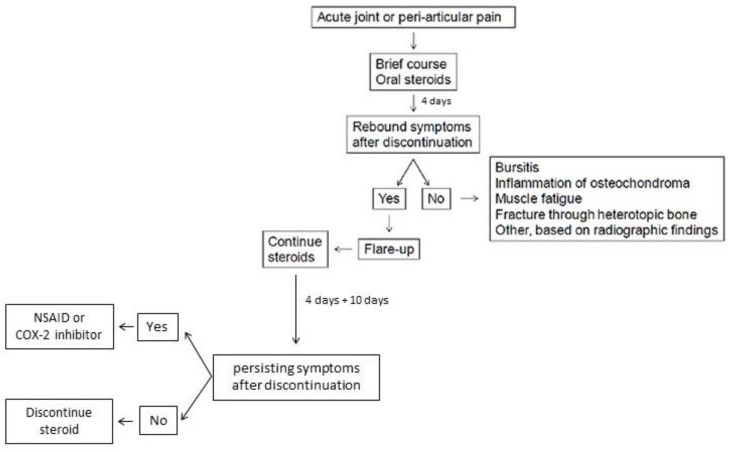
Algorithmic therapeutic approach for flare-ups and acute joint/periarticular pain (from [16], modified).

**Figure 3 genes-12-01187-f003:**
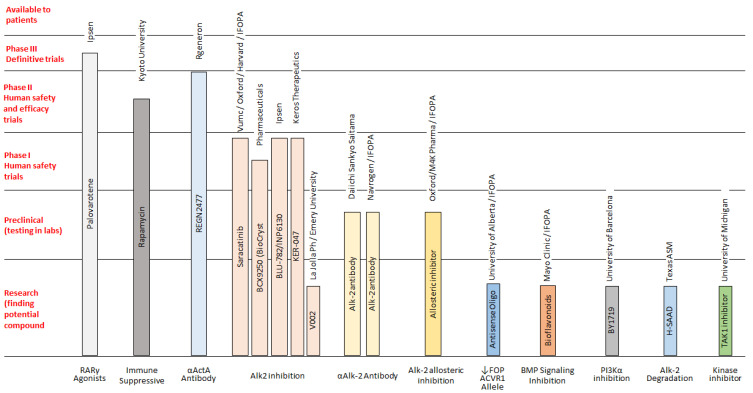
FOP therapeutic pipeline: drugs in research and development [15,16,17,18,19,20,21,22,23,24,25].

**Figure 4 genes-12-01187-f004:**
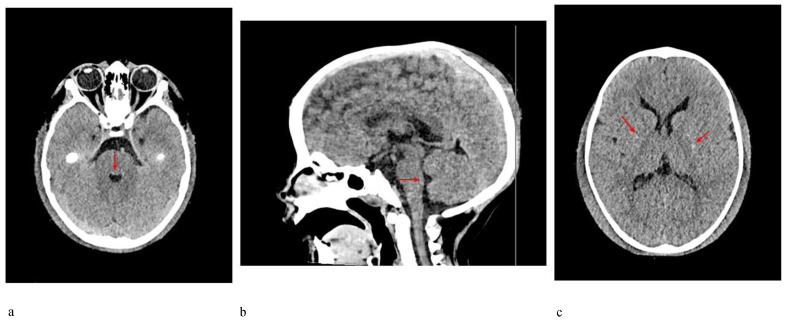
CT scan of the brain of patient 2 showing (**a**,**b**) dysmorphism of the pons with bulging of the dorsal surface (arrows), (**c**) small areas of mineralization at the level of lenticular nuclei (arrows); pericranial soft tissue swelling and edema surrounding cranial bone are also evident in all 3 images.

**Figure 5 genes-12-01187-f005:**
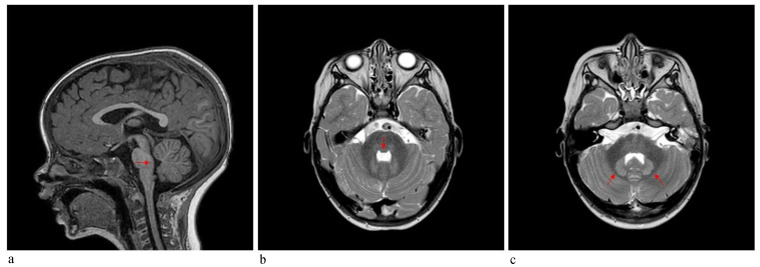
MRI of the brain of patient 2 showing (**a**,**b**) T1 and T2-weighted images confirming dysmorphism of the pons with fusion of the facial colliculi and bulging of the dorsal surface protruding into the fourth ventricle (arrows); (**c**) T2-weighted images show dentate nucleus and peridentate white matter hyperintensities (arrows).

**Figure 6 genes-12-01187-f006:**
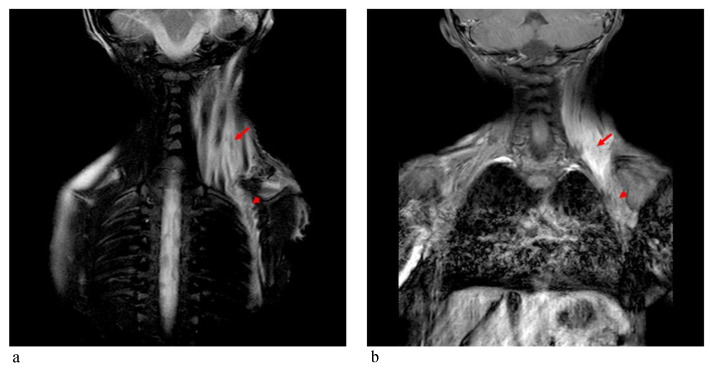
MRI of the cervicothoracic region of patient 2 showing: (**a**) STIR-coronal and (**b**) postcontrast T1 images revealing a diffuse hyperintense signal and contrast enhancement extending along the muscles and fascial planes of the neck (arrow) and thoracic wall (arrowhead).

**Table 1 genes-12-01187-t001:** Classification of FOP, according to clinical features and type of mutation.

	Clinical Findings	*ACVR1/ALK2* Mutation
Classic FOP features	Congenital malformations of the great toes Progressive heterotopic ossification Proximal medial tibial osteochondromas Cervical spine malformations Conductive hearing impairment Hypercalciuria and nephrolithiasis risk Alopecia (spare hair and eyebrows) Neuroimaging abnormalities (especially involving the pontine region)	c.617G>A (p.R206H)
Additional variable FOP features (“FOP plus”)	Short, broad femoral necks Malformations of the thumb	c.617G>A (p.R206H) c.619C>G (p.Q207E)
FOP variants	Normal or minimal changes in great toes Childhood glaucoma Marfan phenotype Cryptorchidism	c.617G>A (p.R206H) c.619C>G (p.Q207E) c.605G>T (p.R202I) c.1124G>C (p.R375P)
	Thumb malformations Cognitive impairment Diffuse scalp thinning	c.774G>C (p.R258S) c. 774G>T (p.R258G)
	Delayed-onset HO Absence of characteristic great toe malformations Severe digital malformations	c.774G>C (p.R258S) c. 974G>C (p.G325A)
	Much more severe malformations of the toes than the classical form Limited motion of the shoulders, neck, chest, elbows, hips, and interphalangeal joints	c.1067G>A (p.G356D)

## Data Availability

The data presented in this study are available on request from the corresponding author. The data are not publicly available due to privacy.

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
