# Peer review of "Fibrodysplasia Ossificans Progressiva: A Challenging Diagnosis"

_genes, 2021, doi:10.3390/genes12081187_

Round 1

Reviewer 1 Report

In this well written manuscript, the authors provide an attentive and cogent review of the main clinical and biological aspects and traits of FOP. They correctly emphasize the critical need to diagnose new patients as soon as possible and thus avoid biopsies or other interventions that often exacerbate the condition and in particular heterotopic ossification. The authors provide a succinct description of several criteria for the correct diagnosis of FOP as well as a description of temporary remedies such as anti-inflammatory drugs that can help against pain or flare-ups. They also mention new drugs against HO several of which are in clinical trials at present. Finally, the authors provide a vivid rendering of diagnostic difficulties and challenges often encountered in the clinics by describing two cases of a 2 year-old patient and a 3 year-old patient and the useful steps and processes that eventually led to the correct diagnosis of FOP. In sum, this review article will be helpful to both practicing clinicians and research scientists to learn about the basics of FOP and useful criteria to diagnose new patients at the youngest age possible, thus avoiding complications and problems of delayed diagnosis and in turn charting a correct management and monitoring of disease progression.

Very minor issues:

Though the manuscript is well written, there are a few grammatical and syntactical errors that need to be corrected.

In Fig. 3 and in the main text, the authors need to cite all the research studies that are responsible for the creation of new possible treatments against FOP. This would also give an opportunity to readers to learn more about how each of these potential treatments came about and what each aims to counter during HO pathogenesis.    

Reviewer 2 Report

The manuscript describes FOP patients carrying ALK2 R206H mutation. FOP is an ultra rare disease of the skeletal dysplasia. But, there are already many case reports on the patients with ALK2 R206H mutation worldwide. I could not find any new information in the FOP research field. At least, I think the following points need improvement.

1) I feel that relationship between information of new therapeutic drugs and result of manuscript is unclear. I think it is better to writing the manuscript  focusing on the diagnosis of FOP.

2) Patient 2 was diagnosed macrocephaly. The relationship between ALK2 mutations and brain tumor (DIPG) has already been reported (Nature Genetics, 2014).  It will be important to discuss this point.

Reviewer 3 Report

The paper describes two clinical cases of FIbrodisplasia Ossificant Progressiva (FOP) summarizing the gold stand signs found in both cases as well as some unusual findings that can help other clinicians to discover new cases. Early diagnosis of FOP is critical to avoid treatments, tests and other conditions that can provoke a new flare-up and bone growth. Therefore, the discover of new symptoms that can be associated to this disease is interesting for both Scientific community and clinicians. Although most of the findings of this paper has been already describes, other symptoms have never been associated to the disease, and can help clinicians to guide future diagnosis.

In general, the paper is well structured and only some minor revisions should be included:

  • In line 47, a reference is missing. 
  • In line 64 to 67 authors say that ACVR1 WT does not respond to activins. Specify that activins block ACVR1 WT signaling.
  • In line 170 authors mention new treatments of FOP that are then included in Figure 3. Please, give more information about these new therapeutical options.
  • Resolution of Figure 2 is low. Please, Improve the resolution of Figure 2.
  • In line 220, please define CRP.
  • In line 132, please define MRI
  • For both patients, please specify how samples for sequencing were obtained.
  • In line 270, please define FNAB.

Round 2

Reviewer 2 Report

The authors have significantly revised the text. Then, I feel that this revised manuscript will be useful for diagnosis and therapy of FOP.